# Information from Earth Observation for the Management of Sustainable Land Use and Land Cover in Brazil: An Analysis of User Needs

**Mercio Cerbaro [1,]\*** [ID]**, Stephen Morse [1], Richard Murphy [1], Jim Lynch [1] and Geoffrey Griffiths [2]**

[1]  Centre for Environment & Sustainability, University of Surrey, Guildford GU2 7XH, UK;
    s.morse@surrey.ac.uk (S.M.); rj.murphy@surrey.ac.uk (R.M.); j.lynch@surrey.ac.uk (J.L.)
[2]  Department of Geography, University of Reading, Reading RG6 6ABRG6 6AB, UK;
    g.h.griffiths@reading.ac.uk
\*  Correspondence: m.cerbaro@surrey.ac.uk

**Abstract:** Brazil has some of the world's most important forest and natural ecosystem resources and their sustainability is of global importance. The expansion of agriculture for livestock, the extractive industries, illegal logging, land conflicts, fire and deforestation are pressures on land use and drivers of land use change in many regions of Brazil. While different institutions in Brazil have sought to use Earth Observation (EO) data to support better land use management and conservation projects, several problems remain at the national and state level in the implementation of EO to support environmental policies and services provided to Brazilian society. This paper presents the results of a systematic analysis of the key challenges in using EO data in land management in Brazil and summarises them in a conceptual model of the factors influencing EO data use for assessing sustainable land use and land cover in Brazil. The research was based on a series of in-depth, semi-structured interviews (43) and structured interviews (53) with key stakeholders who make use of EO data across different locations in Brazil. The major challenges identified in the complex and multifaceted aspects of using this information were associated with access to, and with the processing of, raw data into usable information. The analysis also revealed novel insights on a lack of inter-institutional communication, adequate office infrastructure and personnel, availability of the right type of EO data and funding restrictions, political instability and bureaucracy as factors that limit more effective use of EO data in Brazil at present. We close this analysis by considering how EO information for the sustainable management of land use and land cover can assist institutions as they respond to the varied political and economic instabilities affecting environmental governance and deforestation levels.

**Keywords:** earth observation; indicators; land management; land use; sustainable development; Brazil

## 1. Introduction

Earth Observation (EO) is the process of gathering information about an object from a distance and uses a sensor, usually located on a satellite, to record information about a point of interest [1]. Some EO data and products are available free to the user, from agencies such as the United States Geological Survey (USGS), The National Aeronautics and Space Administration (NASA), and the Brazilian Institute for Space Research (INPE). EO data are used in a wide range of applications, including monitoring natural resources, weather forecasting, agriculture and land management [2]. The free access to EO datasets since Landsat became available to users in 2008 has also enabled different institutions to use these resources to generate new environmental information [3]. For example, EO data from the China–Brazil Earth Resources Satellite (CBERS) and other EO datasets are available at no cost to the end-user via the INPE website [4].

The European Union (EU) Copernicus program also provides free access to global EO data with some key advantages compared with the USGS Landsat program, most notably a spatial resolution (panchromatic) of 10 m for Sentinel 2 satellites (vs. 30 m for Landsat) and revisit times of 5 days (vs. 16 days for Landsat) [3]. Private EO data, provided at no cost for the end-user, are also available via the INPE website, which includes data derived from INPE's contract with UK-DMC2 (Disaster Monitor Constellation) in 2012, where an open license for the end-user was agreed to enhance public monitoring of forest management in Brazil [5]. In Brazil, a country with significant challenges regarding the sustainable management of its natural resources, it has been said that freely available EO data are very important to support different applications and to provide information for end-users [6].

The deforestation monitoring programmes in Brazil rely on such freely available EO data from Landsat to quantify the complete conversion of old-growth forest into agriculture [7]. Since 1988, the Brazilian government has performed annual monitoring with Landsat EO data via the Amazon Deforestation Monitoring Project (PRODES) implemented by INPE, this being a globally-leading example of operational monitoring for quantifying land cover change via EO data [8–10]. Government officials, land managers, researchers, and civil society groups can use information derived from EO data to make better decisions for land use management [9]. The information derived from EO data is used to create detailed maps about deforestation, estimate cropland and pasture development and to assess land use change in tropical areas. Many remote-sensing systems have now been developed to offer a wide range of spatial, spectral and temporal parameters for the needs of different EO data users and organizations [11]. However, delivering products and information services from EO data is far from trivial, as space-borne satellite instruments deliver raw data and users require a wide range of tools and skills to transform this into specific, purposeful information [12]. Better understanding of the processes impacting this transformation from raw data into useful information is something of a gap in the current literature, a gap that the research presented in this paper seeks to address.

Processing large quantities of EO data requires computational resources. Not all potential or actual users have access to such resources nor the training and knowledge to process EO data into useful information [13]. The effort required to convert raw EO data into information (e.g., maps, indicators, other visualisations) for people unfamiliar with the technicalities of EO remains a major challenge for the EO community in building analytical platform applications tailored to various user needs [13]. The Global Earth Observation Systems of Systems (GEOSS) is a good example of a cloud-based infrastructure that manages multiple EO catalogues and provides services via a platform tool that aims to support end-users [14]. However, as noted by Latham et al. [15] "very often users do not distinguish between data and information and they assume what is downloadable is adequate and satisfies their needs". However, potential users span a very wide spectrum, from the public at one end to managers at the other. In between these are specialists working on the processes to transform raw EO data into information. Users (e.g., farmers, researchers, environmental consultants, politicians, civil servants) also have different requirements for the information derived from EO data.

The first analysis of users of EO data and the description of the benefits of the free data policy of CBERS EO data for a wide range of institutions was published in 2007 [16]. Epiphanio [16] showed that the success of free data is proved by the high number of image downloads by users in different regions of Brazil. However, counting image downloads does not necessarily equate to effective usage by public organizations such as the Brazilian Institute of Environment and Renewable Natural Resources (IBAMA), Brazilian Agriculture Research Cooperation (EMBRAPA) and other organizations at the federal level. The second analysis of the user's profile of CBERS EO data was a study based on a sample of 205 participants out of 31,515 people who registered to download the data [17]. Silva et al. [17] showed that 45% of the users had an undergraduate degree, 23% had a master's level degree, 88% a PhD, 22% post-doctoral training, 18% had specific training in EO and 5% had no formal advanced education. The key message emerging from this user survey was that providers of EO data should research end-user needs so as to improve services, understand demand for new products and increase

the quality of the services provided for different applications of EO data. However, few studies have explored the challenges that users of EO face in accessing and applying EO data.

To our knowledge, there have been few published studies on the use of EO for land use management in Brazil which included interviews with end-users, scientists and providers of EO [18–20]. This is despite INPE's pioneering activity on deforestation assessments using EO data since the 1970s [21]. In one of the few studies to date, Rajao [20] reported the outcomes of 85 semi-structured interviews undertaken between June 2007 and August 2009, which included policymakers in Brasilia, senior researchers at INPE and end-users of EO data in different state institutions in the state of Mato Grosso with the aim of collecting information on policy-making and law enforcement practices associated with the use of EO data. The study included the perception and understanding of decision makers on the use of EO data and products derived from EO data provided by INPE to monitor deforestation. More recently, Monteiro and Rajao [18] examined INPE's use of improved EO processing methods to deliver more reliable information to users (e.g., avoidance of errors when classifying fires, deforestation alerts). However, there remains a lack of research on the challenges users confront when trying to access, use and process EO data and little systematic work has been undertaken on analysing user needs. The research reported here addresses this gap by exploring the needs of a wide range of actors in the path from EO data to information in order to identify the major challenges that impact the wider use of EO in supporting the management of sustainable land use and land cover (LULC) in Brazil.

## 2. Materials and Methods

The design of the main elements of the research was based on a preliminary (10 day) study which spanned INPE, EMBRAPA and other institutions in the state of Sao Paulo (e.g., University of Sao Paulo, EMBRAPA Informatics Campinas, Geoflorestas and ArcPlan) in 2016. The preliminary study was used to plan the main phases of fieldwork in terms of questions that needed to be asked and potential stakeholders who should be approached. It was recommended that a more 'open' or 'grounded' style should be adopted as far as possible so as to allow for the emergence of insights regarding the use of EO data in Brazil.

Following on from the preliminary study in 2016, the main fieldwork comprised a series of interviews with key stakeholders in federal and state institutions in Brazil conducted in two phases in 2017. The main difference between the phases was that Phase 1 focused primarily on institutions at the national (federal) scale, while Phase 2 focused on institutions at the state scale, although given that these interact there was some overlap in terms of the inclusion of federal and state-level actors. The selection of interviewees was based on institutions known to be the most important in the public sector associated with agriculture, forest and environmental management in Brazil and included an element of 'snowballing' whereby interviewees suggested others who should also be interviewed. To protect the confidentiality of the interviewee and to avoid data protection issues with the interview data, all interviewee names have been anonymized. The questions in each of the two phases of the data gathering interviews are given in Appendix A.

### 2.1. Phase 1

This phase of data collection comprised semi-structured interviews with 27 institutions in Brasilia and 16 institutions in other locations (see Appendix B). The 52 h of interviews were recorded (with permission) for post-interview analysis of emergent themes. The interviews in Brasilia with key institutions associated with Forestry, Agriculture and the Environment were used to develop understanding of the challenges and opportunities faced at the federal scale, but this also included their interaction with state-level institutions. All interviewees were asked the same 20 questions in the semi-structured format with scope included for the open raising of points by the interviewees in order to capture emergent insights in a grounded theory approach.

### 2.2. Phase 2

Phase 2 comprised a series of 53 semi-structured interviews in Brasilia, Cuiaba (Capital city of Mato Grosso State) and Rio Branco (Capital city of Acre State) (Appendix C). The interview questions (Appendix A) were informed by the results of the interviews undertaken in Phase 1 and were designed to invite the interviewees to discuss specific challenges faced by their institution. As in Phase 1, scope was provided for interviewees to also raise any issues that were not covered in the questions. The interviews in Phase 2 were not recorded (unlike those of Phase 1) but instead contemporaneous notes were taken by the interviewer during the interview session.

All the interview material of Phases 1 and 2 was subjected to thematic analysis to identify key issues. The issues that emerged from the interviews were grouped and formed the basis of the domain with sub-elements structure of a conceptual model (Figure 1) to represent the system, as perceived by the interviewees. It is recognised that the in-depth interviews each gathered data from one person at one time [22] and, as Walliman [23] points out, they represent a situation in one place, at a particular time as seen by that observer. In appraising the results, we were careful to recognise this attribute of the individual interview responses, but we consider that aggregation of the large number of in-depth interview responses through the analysis supported the creation of the generalised, system-level representation in Figure 1. The model is presented in the following section together with selected quotations to illustrate the main points within each of its domains and elements.

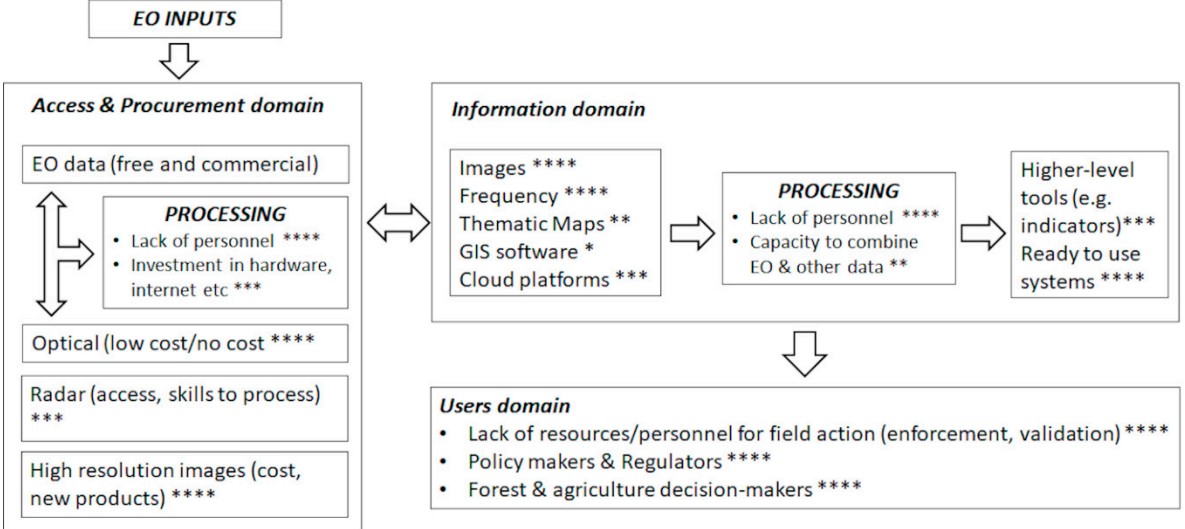

**Figure 1.** Conceptual model developed from the interview responses to illustrate the main steps and limitations in the flows of Earth Observation (EO) data and information for use in the management of land use and land cover (LULC) in Brazil.

## 3. Results

### 3.1. Conceptual Model of Factors Influencing EO Data Use for Assessing Sustainable Land Use and Land Cover in Brazil

The conceptual model of factors influencing EO data use for assessing sustainable land use and land cover in Brazil synthesised from the analysis of the interviews is shown in Figure 1. The model tends towards a greater emphasis on the implementation of existing policy rather than on targetting policy development, although where the latter was mentioned by some interviewees that element has been included. Asterisks in the model indicate how frequently each of the elements was raised in the interviews; the larger the numbers of asterisks the more frequently the point was mentioned. The

model is based largely on responses from public institutions (federal and state agencies) because these are where most of the interviewees were situated. The three domains identified in the model are:

1.　　Access and Procurement (of EO data and products).
2.　　Information (derived from EO data).
3.　　Users (of EO-based information/data).

The 'Access and Procurement' domain is where 'raw' EO data inputs enters the model (top left) and receives primary processing to generate images, maps, GIS databases and data uploaded to cloud platforms (web-based mapping). A large number of interviewees noted the need for further (higher-level) processing to generate what several referred to as "ready to use" systems, including indicators—this activity area recognised by the interviewees is the basis of the 'Information' domain. The final area, mentioned by all interviewees, was 'Users'—this includes the resources necessary to use the information in the field and is represented as the 'Users' domain of the model.

There are many elements (and links) between, and within, the three domains in Figure 1. It was noted also that several of the key elements from the interviews were features of more than one domain (e.g., lack of personnel) whilst others were specific to a single domain. The following results presentation is centred on the domains but a number of common elements will be seen in more than one domain.

*3.2. Access and Procurement Domain: General Optical EO Data Availability and Processing Capacity*

In this domain of Figure 1 under Processing, a lack of personnel was mentioned by all interviewees as the main problem in public institutions. It was also mentioned by all interviewees that the availability of optical data (low cost/no cost) was reasonably good and essential, but that there were issues with radar EO data (access, skills to process) and the need for up-to-date high resolution imagery (see also Section 3.3). Interviewees tended to note that the challenges they faced were less with data availability than with the capacity to process that data. However, this having been said, there were differences identified in the capacity to process EO data across the range of institutions in this study. At institutions such as INPE, National Water Agency (ANA), IBAMA, Brazilian Forest Services and EMBRAPA EO image processing is part of the daily routine of civil servants and there appears to be enough technical personnel with the skills to process and use the EO data available to them. The following illustrate this point:

> *We use EO data to plan land use, to evaluate environmental impacts and for landscape planning. We improved the use of EO data here in Rio Branco and the national level. However, we have too much EO data available and only limited information based on the EO data.* *For example: Terabytes of EO data are available but only limited information based on that EO data. In my view we need to improve people skills so they can extract information from the EO data for the final decision-making process. The end-user needs to know the right question to solve a specific problem and needs to understand the best EO data to solve that problem. For example: soil vulnerability or how to measure forest biomass.*
>
> (Senior director, EMBRAPA, Rio Branco)
>
> *I work in a sector with technical capacity here at INPE, but what does not exist is a logical organization to restructure the sector at the national level. We have a different reality now and the institutional arrangements are different in 2017 than in the past.*
>
> (Senior scientist D from INPE, Sao Jose dos Campos)
>
> *We need to aggregate institutions within our national project, but institutions need to set aside their individuality and work in groups. Sometimes we encounter institutions that cannot work in this way. Sometimes we have public institutions in Brazil working on the same project of other institution because they do not have a joint vision, only a personal view.*

*We need to have a critical analysis to create partnerships and cooperation. Nobody does anything alone. For example: we have a technical cooperation between the National Meteorology Centre (INMET) and the National Food Supply Company (CONAB) to use the modern IT labs at INMET, and in some situations we solve problems together.*

(Senior director, CONAB, Brasilia)

*Between INPE and EMBRAPA we have about 300 people in different groups that are specialists in Earth Observation. EMBRAPA uses EO data for different applications in Agriculture, Forestry and the Environment. We also have different universities at the national level that provide courses and training in Earth Observation and GIS. What we need to improve at the national level in Brazil is institutional cooperation and partnerships between different institutions to share information and help institutions with technical knowledge. It has improved a lot, but we need to improve more.*

(Senior director, EMBRAPA, Brasilia)

However, differences in the availability of skilled personnel and technical capacity were recognised between institutions in the regions and those based in Brasilia. For example:

*The main problem is a lack of knowledge in Earth Observation and availability of workers with technical knowledge to generate information for our managers at the state level. We need to invest in IT (e.g., computers, internet) and training of our personnel to generate the information with the EO data that are available. In Brasilia we have enough personnel with skills in Geographical Information Systems (GIS) and Earth Observation.*

(Strategic analyst A, INCRA, Brasilia)

The head of the Monitoring Division at ICMBIO highlighted some differences in the main demands between the headquarters in Brasilia and regional offices as follows:

*Demands of headquarters: to update GIS software, improve database systems, improve technical capacity of our staff and to improve the quality of the internet.*

*Demands at the Regional offices: to improve technical capacity in geoprocessing, improve the quality of the internet and to improve the reverse flow of information between regional offices and Brasilia. The lack of personnel limits the capacity of regional offices to send the information back to headquarters and this limits our capacity to monitor deforestation. Again, I would like to highlight the very slow internet as one of the main problems at regional offices.*

(Senior manager, ICMBIO, Brasilia)

However, in a country as large as Brazil there will almost inevitably be an issue with capacity:

*I would say the renewal of workers is maybe one of the most serious problems in public organizations in Brazil. It is not enough to have the data and put in the frame. You need to gain knowledge and actions based on that information. EO information brings benefits for society. Now, if you imagine a country with over 5,000 municipalities, the barriers and limitations to the use of data are enormous. The attempt to regulate the information, for one side we have the idea that everyone will use it in the most correct way. In practice, it is very difficult. For example, you go to a distant city and the mayor bought a drone from the internet. Then, he will try to fly over the city to help him to collect information for house taxes purposes (IPTU), to save money rather than hire a specialized enterprise. Someone told the mayor this was the cheapest way to acquire information, but nobody in that city understands geoprocessing, EO data management, georeferencing, and how to use the information. You need to have training, clarification at the local level, consolidated purchases of EO data at the national level and consolidated practices.*

(Senior staff at the Brazilian Space Agency, Brasilia)

*3.3. Access and Procurement Domain: Specialised EO Data (Radar and High-Resolution) Availability and Processing Capacity*

While the availability of general optical EO data was not a major issue, this was not the case with more specialised data such as those from radar and high-resolution instruments (<1 m resolution). Radar systems can penetrate cloud cover unlike optical systems [24] and the lack of availability of radar EO and frequency of EO data was mentioned as one of the main limitations. In addition, over half of the interviewees noted skill shortages in the processing and use of EO data and felt it necessary to have technical training for this in different institutions at the state level.

One of the main limitations is access to radar EO data and EO data with high temporal revisit times, especially for the Amazon region. It was considered a critical limitation when optical EO data are not available due to cloud cover and when end-users do not know how to use radar data (even if such data were available for use). Problems associated with the speed of the internet and investment in IT infrastructure were also mentioned as major challenges to download EO data (optical and radar) in remote locations and in state institutions in Rio Branco and Cuiaba. The following quotations illustrate these points.

> *In 2015 we started to use GOOGLE Earth Engine. I have the script of the Earth Engine to remove cloud pixel by pixel and it gives me the resolution and the geographical area of interest. I apply 50% of my EO data use on Earth Engine for detailed analysis such as time-lapse. The main limitations here in our institution is access to radar images for the Amazon region.*

(Senior manager at Socio-Environmental Institute, Brasilia)

> *We need to improve the speed of the internet, invest in data management to protect our institutional archives and hire more people with geoprocessing skill. I work on my personal laptop when I need to use EO data because the computers available at our institution are too old.*

(Land analyst, Land Institute of Mato Grosso (INTERMAT), Cuiaba)

> *One of the key problems is the thermal infrared and frequency of EO data, including the images of LANDSAT 8. Sometimes we have a restriction on the number of images, for example during the raining seasons in Brasilia. This is a factor that limits the use of information. Technical capacity is not a problem at the National Water Agency in Brasilia. We have different people with masters and PhD in remote sensing. I would say our major limitations are availability of specific EO data. I would say the availability of thermal satellites with more temporal frequency in the raining season.*

(Senior environmental analyst, National Water Agency (ANA), Brasilia)

> *Access to images, there are things that need to improve. BAND L, RADAR. The sensors that could help are RADAR. Today, we do not have a systematic system of BAND L. We use SENTINEL B and C and COSMOS BAND X.*

(Senior officer, Ministry of Defence, Brasilia)

Using EO information to monitor deforestation in the Amazon region was considered essential in order to provide reliable information to guide action on the ground. The frequency of EO coverage, the availability of specialist EO data (e.g., thermal, radar) and the skills to use it were seen as main limitations to meeting these demands of end-users. There are centres of excellence in Brazil that could provide such training e.g., INPE, EMBRAPA and IBAMA. However, the need for better cooperation between ministries and institutions at the state level was mentioned by more than 50% of interviewees as a challenge to the improved exchange of EO-based knowledge.

Several interviewees mentioned the need to have access to high spatial resolution data of 1 m or finer. For example, The Ministry of Environment (MMA) purchased 5 m high resolution data in 2012 for the National Rural Environmental Registry (CAR, Cadastro Ambiental Rural), which is necessary to help inform land use policy at the national level [25]. CAR was an important breakthrough of

the Native Vegetation Protection Law for environmental monitoring in Brazil, and it worked as an important tool to assess where property owners can perform forest management but not clear-cut forests in Legal Reserve Areas (LRs) or Areas of Permanent Preservation (APPs) [25]. The LRs depends on biome, vegetation type and deforestation date [25]. This includes a self-declared land use registry for rural properties where inputs to the system by the end-users would not be possible without the use of high-resolution EO data [25]. However, several interviewees in different institutions reported problems with the validation of the information by environmental authorities as the information was self-declared and not every user inserted land use information correctly.

> *We need EO data with high resolution for the Rural Land Registration (CAR) and high frequency EO data for the Amazon region. It would be great to have both data available for public access via INPE.*
>
> (Environmental project analyst, Brazilian Forest Service, Brasilia)

> *Now, we are proposing the acquisition of images of high resolution for small properties [agriculture land areas between 1 and 2 hectares] of 1 m resolution or better. They need to be part of the Rural Land Registration (CAR). The images of Rapideye 5m resolution is not enough and do not address all our needs. We had a first vision of the entire territory with 5m. Now, we will need a specific view of those small areas with high resolution data. This will provide better information for crop estimation and to identify all the agricultural production areas; 70% of the food supply in Brazil comes from family farming in small properties.*
>
> (Senior director B, IBAMA, Brasilia)

High resolution EO data were therefore deemed essential to implement the new phase of the National Rural Environmental Registry (CAR), which has been expanded to include relatively small areas of land between 1 and 5 hectare (ha). This and other demands for such information has grown considerably in recent years:

> *The demand for EO data by the Brazilian state has grown exponentially in the last few years. A few decades ago, we just had a few organizations asking for EO data from INPE. Today, we have the Ministry of Cities, Ministry of Environment, Ministry of Defence, and Ministry of Planning. Basically, there is no ministry, secretaries, groups, or a project that at one point do not use information from EO data to a certain level. This not only at the federal but state and municipal level as well. Systems to control buses, control taxes, land use regulation at local level. The demand is explosive to a point that is difficult for any organization in Brazil to understand how much the Brazilian state demands in terms of EO information. Recently, I am saying about 3 years ago, different Ministries decided to start a consolidated purchase, such as the National Rural Environmental Registry (CAR).*
>
> (Senior staff at the Brazilian Space Agency, Brasilia)

Senior staff at the Brazilian Space Agency (BSA) noted the challenges involved in measuring how much EO-based information is needed at the federal and regional levels. One issue here seems to be the valuation of EO data by those with the power to make decisions:

> *I think there is a lack of connection between the image alone and the use. There are applications. For example, a mobile phone access to facilitate work in real time. We still have a barrier. For example, I do not see all city councils prepared to use the information provided not even from the national rural registration (CAR). We need someone that understands the use of satellite images to make a product for someone to take a decision based on information, and you need that person who have the power to take the decision to value that information as essential, we still have this issue. Things are still in the political arena.*
>
> (Senior staff at the Brazilian Forest Services, Brasilia)

### 3.4. Information Domain: EO Data to Information

In the Information Domain of Figure 1, interviewees mentioned that access to images, frequency and cloud-based IT platforms are important to manage the volume of EO data and to facilitate selection of the right EO data for the end-user. Everyone mentioned the need to move to cloud-based platforms, GIS systems and thematic maps as the basis for "ready to use systems" for users. It was suggested that new platforms could reduce the problem of downloading large EO datasets given the relatively poor speed of internet in institutions at the state level. The challenges in the Information domain are often associated with lack of higher tools (e.g., ready to use tools or indicators) to facilitate the use of information. Ready-to-use tools could reduce the challenges institutions face to organize, store, manage and analyse different types of EO data. However, lack of personnel was mentioned as one of the main issues for two reasons, (i) when generating EO-based information and (ii) to work on the ground (e.g., IBAMA law enforcement agents, ICMBIO conservation areas, firefighters). A point which was often mentioned in the interviews is the need to go beyond data and provide 'information' that policy makers and managers can work with. The following illustrates this:

*The minister will not know all the technical GIS and EO terms. The minister needs to know what type of information is available and the potential to use the information generated from EO data and GIS. What I noticed related to information derived from EO data to support environmental policies and decisions when I worked at IBAMA are the challenges to implement the use of new information within public institutions.*

(Infrastructure Analyst, Ministry of Mines and Energy, Brasilia)

This same point emerged in a number of ways from interviewees, including a lack of translation from EO data into information in a form that managers, and indeed policy makers, can use. For example, as stated by a senior policymaker at the Ministry of Environment:

*We are at the end of the decision process and we identify the type of information we want to receive. So, it is not a spontaneous action. The information can be important, and we listen to the people in science, but to assimilate the information is a second stage. The most important factor for me is the consistency of information and that all the land use classifications are well defined in the territory. The second factor is the use of information to influence public policies. For example: all the information of fires at the national level are derived via EO data and we need to establish a national public policy based on the information available. Sometimes EO information is not enough to promote a public policy and sometimes the Brazilian congress does not approve the legislation. Conservation, biodiversity, deforestation, mitigation of greenhouse gas. There are different views inside the government. The solution for public policies sometimes is not at the technical level, and at the time of implementation we have different views of development.*

(Senior policy maker B at the Ministry of Environment, Brasilia)

*The main challenge is to share knowledge even if we have been working for over 20 years in this area of land use and the protection of indigenous land. People in government do not understand the importance of geospatial information and how this influences the decision-making process. For example, one politician cannot have this dimension, and this is not taking to a level of decision making to increase the quality of the use of information. Although I have good technicians and we need to develop good products for this information to arrive at the president of FUNAI, Ministry of Justice and the Ministry of Lower House. Politicians cannot realize the importance of this process and how this can change people lives.*

(General Coordinator A, FUNAI, Brasilia)

*"The key question to answer your question between the main limitations on the use of EO data in Brazil is the link between EO data providers and EO data users. The technical capacity of end-users*

*to download EO data, process the data and use the information derived from satellites for different applications. The parallel, if you imagine a situation where the model of providing EO data to users has always been a model where EO data was generated by a particular satellite with different sensors and provided to users by space agencies. The EO agency received the EO data and then provided this EO data by selling or by providing EO data at no cost for end-users. However, in this case, a particular group of users received the EO data and processed the data into usable information for specific applications. This model generated good results in Brazil, overseas and it encouraged more people to use EO data. In the past, EO data was expensive and difficult to buy, and then when EO data were available for end-users. EO data are relatively easy to acquire with limitations for high resolution data. The bottleneck of access was solved, then the most important limitations was those who interpreted the EO data. Now, the problem is processing EO data, intellectual capacity of users, personnel available in institutions, and the capacity of institutions to manage big EO data and all the information derived from EO data.*

(Senior Researcher A at INPE, Sao Jose dos Campos)

*EO should provide complementary information to plan different public policies. For example, it complements the information of soil sciences to plan a national soil policy but does not define the best agriculture practices for different geographical regions. We need to integrate with different data to plan public policies. It is difficult to use EO alone to define policies and strategies to improve agricultural production.*

(Senior director IBAMA, Brasilia)

Indeed, for one interviewee it is almost as if there can be too much information and this needs further "elaboration" to help identify the key messages:

*Having information is not enough, we need to improve the quality of this information. We increased the quantity of EO data, yes. We increased the number of users of EO data, yes. But sometimes I feel that additional detailed information is needed. For example: We monitor vegetation cover of forests at the national level in Brazil, but we are always looking at the state of the forests. Sometimes we are mapping and detecting new deforestation areas, but sometimes one forest is very degraded, another one has a high level of biodiversity, and it has not been damaged by humans. We need to quantify and improve the quality of the information.*

(Senior manager Brazilian Forest Service, Brasilia)

However, such translation from processed data to information, even if only in a basic map form, requires additional expertise. For example:

*EO data are not a problem and we use a lot of data; the problem we have here at ICMBIO is the institutional capacity to use and process all the data available. The focus here is on units of conservation. As you can see, our team is very small. We have interns and two more interns that are not here today. We do not have the capacity to generate all the information and we use products from INPE. All the data about deforestation and fire is from INPE. We produce new information based on INPE data for the areas that were affected by fire in units of conservation.*

(Senior manager ICMBIO, Brasilia)

Thus, according to this interviewee, the quantity and type of EO data available are not in themselves a limiting factor; it is the capacity needed to go from data to information that people can use which is the issue. National institutions such as IBAMA and EMBRAPA are capable of processing advanced EO data into information, but new institutional arrangements and cooperation are necessary to transfer that expertise to other institutions. The emerging cloud computing technologies provide new opportunities to optimize the use of existing EO data into value-added products and information [26]. Online tool such as Google Earth Engine also allow users to perform EO data analyses [27] and empower the use of EO data by a wider audience that lacks technical capacity in EO

data processing [28]. Other private providers such as the Amazon Web Services and Planet Labs also provide cloud computing capabilities for mapping and monitoring actions [29].

There was something of a tendency for interviewees to blame other institutions for lack of cooperation and lack of willingness to establish cooperation and partnerships and this is by no means a new observation. Rajão [20] illustrated the blame-avoidance of institutions associated with environmental crises in Brazil, factors that underline much of political, institutional behaviour in practice and between bureaucratic actors. Interestingly, one interviewee went further than the above and suggested that there was an "aversion of information" in government which limited the use of EO-derived data:

> *I do not think there is a lack of competency within institutions. The problem is a lack of political will and means for exercising the competences and coordination at the national level. For example, the challenges to have different ministries to sign the same goal is almost an attempt of despair. For me, the explanation is based on politics. The nature of politics. It is not only generating the information but keeping and maintaining the time series basically is a key instrument to long-term planning in Brazil. To understand why culturally and politically we have challenges. For example, the decision of not producing a new agricultural census. Yes, we have less money available now, but I consider historically the question to generate new data a second plan for the government. This hidden thinking in government. I think there is an aversion to information. It is intentional not to have the information. This results from what is happening in the Brazilian state, levels of corruption, endogamy, for me the problems with the use of EO data are the same problems of corruption and the forms of political representation. This is not lacking a better "clearing house", or allocation of more financial resources deliberately, or what can we do to better to use the EO data. If someone does not want to have the information and is not willing to use the data, how are you going to fight against this? This is not only on remote sensing and EO data, this problem happens on the generation of basic information to support decision making in the state of Brazil.*

(Senior scientist B at the Ministry of Environment, Brasilia)

This aversion to information in government was only mentioned by one interviewee but is intriguing nonetheless and does resonate with a more recent event with regard to the President of Brazil accusing INPE of lying about the figures derived from EO data showing an increase of deforestation rates in Brazil, this leading to the firing of the agency's Director on 2nd August 2019 [30].

*3.5. Users Domain: Capacity to Act on the Ground*

The final factor which was noted by all interviewees was the lack of capacity to act on the EO-based information that the agencies receive. For example, ICMBIO in Acre noted how a lack of funding prevented them from sending teams to the field to verify the EO information provided from Brasilia. Therefore, it is not only a matter of the capacity to generate information but also having the capacity to assimilate and make use of that information, as noted by several interviewees at the Brazilian Institute of the Environment and Renewable Natural Resources (IBAMA), ICMBIO, and the Ministry of Defence:

> *We cannot monitor all the polygons identified in the EO data. The cost of mobility to act on deforestation in all regions is high. We send only the most important polygons of deforestation for our teams in the field. We have rules for that based on priorities in key geographical areas. The scenario to visit all the polygons of deforestation in the field is unattainable. If instead of 2000 staff in the field, we added 20,000 staff would this reduce deforestation? Maybe not. Then, it is difficult for me to say how to solve the problem of law enforcement of deforestation or to reduce deforestation to zero; do we need 20,000 staff with knowledge in EO processing and remote sensing? Maybe not. We have other factors influencing use. There are economic factors. People are there living in the Amazon region and they need to produce. We have a scenario, for example, illegal timber is more profitable than legal wood based on certified forest management (3–5 years management plan).*

(Senior staff A, IBAMA, Brasilia)

*The use of information, this is another problem, when the information is used for law enforcement of deforestation areas in the Amazon region. The information is so rich that law enforcement is not prepared/able to execute and use all the EO information available. We deliver 100,000 polygons annually and IBAMA provided the evidence that if they check 1,000 polygons it is too much, IBAMA is visiting less than that on the field in the Amazon region.*

(Senior scientist B, INPE, Sao Jose dos Campos)

*The political and economic context affects ICMBIO. We have had financial cuts of 40%. We need more people, additional money for gasoline to send the teams in the field. We also need to improve the speed of internet and new investments in technology (computers and data management). The demands are linked to the capacity of the government to invest in ICMBIO in Brasilia and here in the state of Acre. For example: we have enough cars, but we do not have the financial resources to visit all the locations.*

(Environmental Analyst, ICMBIO, Rio Branco)

*If you use the army, marines or aeronautics, you need to pay for the costs and fuel. The helicopter needs maintenance after certain hours of flight. Another question is the problem of monitoring. For example, access to SENTINEL data could increase our power to monitor, but not the power to act. We have resources to detect, but we do not have money to act in the field. The satellite information is just information. Then, it depends on what are you going to do with this information. We can detect deforestation, expansion of agriculture, livestock. The key decision is how to act based on this information.*

(Senior officer, Ministry of Defence, Brasilia)

This ability, or indeed inability, to act on the ground using information derived from EO data often receives rather little emphasis in studies designed to explore the use of EO data. It can be compounded by user errors in selecting, processing and analysing EO data which can result in inappropriate recommendations and decisions [27]. There is clearly a demand for EO tools to support the progression from raw data to action so it is perhaps surprising that, to date, the focus of most studies has been largely on the availability of EO data and their translation into images, maps, etc. In the present work the use of these outputs as information for users and for associated actions was clearly articulated by the interviewees and this points to a desire for further investment in personnel to work at the interface between use of the information derived from EO data and action in the field.

## 4. Discussion

This research is the first to take a broad perspective on the challenges involved in applying EO to help/facilitate sustainable management of LULC in Brazil. Interviewees were from the most relevant public institutions associated with land use management in Brasilia and at the state level in Mato Grosso and Acre. The cities of Cuiaba and Rio Branco were selected at the state level to explore the challenges involved in applying EO at the state level, as well to understand the challenges between the capital Brasilia and institutions at the state level.

The design and conduct of the semi-structured interviews provided full scope for the interviewees to raise any issue they considered important and the conceptual model in Figure 1 is a high-level integration of the main elements to emerge from those interview responses. As discussed below, the model raises several points that resonate with the wider literature e.g., skills, although several new observations are made. For example, it is abundantly clear that the challenges to the wider adoption of EO-based information are complex, often related to financial limitations and pressures and that solutions will benefit from enhanced inter-institutional cooperation. Given the turbulent economic and political nature of Brazil at the time of the interviews and since, this is unsurprising. The results presented here are in line with those of Monteiro and Rajao [18] who reviewed the problems of policy-science interface in Brazil faced by scientists and the different ways government institutions interpret EO data to promote reginal land use planning. However, the findings of this research show

that there are various limitations to the use of EO derived data for LULC in the respondent institutions and these not only operate at the interface between science and policymakers but are inter-related through complex interactions. The results of our work differ from Monteiro and Rajao [18] by focussing on several limitations end-users are confronting when using EO data, and not only on how scientists at INPE deal with tensions emerging from their roles as providers of information and as citizens concerned with how their research influences policy and politics in Brazil. We have examined the limitations associated with end-users of EO data and covered a wide range of issues, and not only on aspects associated with the use of EO data and deforestation.

Interviewees made a clear distinction between the different capacity of different users to be able to access and handle different types of EO data. Expertise to process and use radar EO data was available at the federal level (Brasilia) but was much less available at regional levels, and indeed the latter was noted as a limitation by several interviewees. A lack of personnel was noted as a major problem in public institutions and is reflected in the model's Access and Procurement domain. It was also mentioned universally that the availability of optical data (low cost/no cost) is essential and reasonably good, but that issues existed with radar EO data (access, skills needed to process the data). Radar EO data with high coverage frequency was considered important/essential to monitor deforestation during the cloudy months in the Amazon region but access and sufficient ability to process this data was lacking despite Sentinel 1 EO radar data being available at no cost. IBAMA and EMBRAPA have personnel specialised in working with EO radar data but no technical cooperation was in place during the time of the interviews to allow that expertise to be shared. A need was also widely recognised for up-to-date high-resolution optical imagery to monitor, for example, land use in small properties at the national level. Over half the interviewees commented that the Ministry of Environment (MMA) should purchase new high-resolution data for land use mapping at the national level and provide that information at no-cost for end-users via INPE.

In the model's Information domain (Figure 1), interviewees mentioned that access to images, frequency and cloud-based data/information platforms to help address the volume of EO data are important to facilitate the selection and provision of the right EO data by the end-user. Everyone mentioned the need to move to cloud-based platforms, GIS systems and thematic maps as the basis for 'ready to use systems' for users. It was suggested that new platforms could reduce the problem of downloading large EO datasets given the relatively poor speed of the internet in institutions at the state level. The capacity of individual users to process and select large EO archives emerged as one of the major challenges. This need for tools so end-users can process and handle large volumes of data has been noted previously [31]. Indeed, the wealth of information available in EO data archives (e.g., from Landsat) presents challenges for their integration and analysis which has meant that these data have yet to be fully exploited [32].

Grainger [33] outlined 12 components from sensor design and launching of a satellite to enable delivery of usable information and knowledge to end-users. However, not all EO data are converted into usable information, nor do the information products derived from EO data necessarily meet user needs. To help overcome this problem, high-performance computing power (e.g., cloud platforms to process EO data) was considered by interviewees to offer more user-focussed ability to integrate and rapidly process environmental and EO data over their full spatial and temporal dimensions as also noted in the work of Lewis et al. [32]. However, Camara et al. [34] have outlined the major challenges facing researchers when designing new solutions based on data analytics technology and these include the problem of designing EO data analytics that meet the needs of users—these being particularly relevant to the Information domain of the model in the present research (Figure 1).

New geospatial web services (e.g., web-based mapping applications) are needed to process large volume of EO data with different spectral, temporal and spatial resolution stored at different places into useful information to help meet users needs (e.g., simple interface that can be used with a few commands) [35]. The most used MapReduce-based tool for processing large volume of EO data is the Google Earth Engine (GEE) [35]. Another example is the Australian Geoscience Data Cube (AGDC),

an infrastructure that uses high performance data analytics to process large volume of EO data and aims to improve the use of EO information by governments, scientists and other users [36]. Given that it was clear from the results reported here that users have different technical capabilities to process and use EO data, then it is perhaps understandable that some interviewees emphasised "ready to use" systems such as the AGDC. Converting EO data into usable information at a large scale is an organizational challenge that requires different stages to transform data into usable information [37]. When comparing the results of our research with those of Rajao [20], the present work adds important new perspectives on the specific institutional challenges of transforming raw EO data into usable information; a point which emerged strongly in the responses of our interviewees.

Finally, all interviewees made a clear link between the development of the EO information and action in the field and noted that the provision of adequate personnel and equipment were often limiting at that point. Indeed, despite the constraints and enablers noted in the Access and Procurement and the Information domains of Figure 1, the main limitation which emerged is not a lack of EO derived information per se, but funding and institutional capacity for field action to make use of that information, even if that information can be made available in highly "ready to use" form (still something of an important limitation). For example, although senior directors at IBAMA and BSF noted that the emergence of high-efficiency EO products with advanced processing capabilities will reduce operational costs and support decision-making in several institutions, they acknowledged that the key limiting factor will still be action at the field level. The tensions in the User domain because of limitations in linking EO information into actions on the ground indicate a need for greater analysis of this domain. For example, the senior coordinator at the National Indigenous Foundation (FUNAI) in Brasilia mentioned that maps and EO information have been used for the last 20 years, but the main challenge is how ministers and high-level politicians are using the information to promote sustainable development in the Amazon region.

The combination of information derived from EO data with other sources such as socio-economic, biophysical or topographical data was mentioned as essential to provide solutions. Much can be done to improve the flow of EO data and information (provided resources are available) within the Access and Procurement and the Information domains of Figure 1, areas of EO 'technology and processing' that have often received the most emphasis in analyses of EO for LULC. The work of the present study, however, has also highlighted that lack of resources and policy conflicts within the Users domain act as major constraints on actions that these users could be taking based on valuable information increasingly becoming available through the emerging "ready to use" EO systems.

## 5. Conclusions

This research has identified the following limitations to, and opportunities for, a more effective use of EO in land management in Brazil:

- Improve technical cooperation between institutions.
- New 'ready to use products' as an important element for the decision-making process.
- There is a specific lack of technical skills to analyse and process radar and high-resolution optical EO data.
- New investments in IT infrastructure and fast internet in public institutions at the state level.
- Need for investment at the field level action (law enforcement and validation of information on the ground) to help overcome the main barriers to 'use' even when EO information is available. More research is required to explore the reasons preventing investment at this level and how the actors at this level can make better use of 'ready to use' systems
- EO derived data are essential for monitoring deforestation in the Amazon region and for supporting the role of CAR. The implementation of CAR from state regulation to federal law and policy requires further investigation, especially with regard to the challenges faced by government institutions when implement CAR at the national level.

Our work suggests that improved understanding from ongoing research with users and providers is going to be as important as the ongoing advances in EO technology and in policy-making as we seek to optimise the impact of EO information for the sustainable management of LULC.

**Author Contributions:** Conceptualization, M.C., S.M., R.M., G.G. and J.L.; methodology, M.C., S.M., R.M., G.G. and J.L.; formal analysis, M.C., S.M., R.M.; investigation, M.C.; resources, M.C., S.M., R.M., G.G. and J.L.; data curation, M.C.; writing—original draft preparation, M.C.; writing—review and editing, M.C., S.M., R.M., G.G. and J.L.; supervision, S.M., R.M., G.G. and J.L., project administration, S.M.; funding acquisition, S.M., R.M., G.G. and J.L. All authors have read and agreed to the published version of the manuscript.

**Funding:** The first author's PhD study programme at the Unversity of Surrey, UK was funded by the Natural Environment Research Council's (NERC) SCENARIO Doctoral Training Partnership (NE/L002566/1) which is gratefully acknowledged.

**Acknowledgments:** We are most grateful to Gilberto Camara and INPE for their assistance in the conduct of the research reported in this paper. We would also like to thank all the institutions and individuals who kindly participated in the research for their time and cooperation.

**Conflicts of Interest:** The authors declare no conflict of interest.

## Appendix A

**Table A1.** Interview questions asked in Phase 1 and 2 semi-structured interviews.

| Phase 1 (20 Questions) |
| --- |
| What type of organization? |
| What kind of role do Earth Observation play in your current activity? |
| Please give examples of how your activity benefits from EO data. What are the main uses and needs of EO data in your activity? |
| In which geographic area is your organization active? Who are the main users/customers of EO information? |
| In your view, what are the main needs to provide specific environmental services in different regions of the country? |
| What type of EO data does your organization usually use? |
| In your view, what is limiting the use of EO? |
| In your opinion, what are the main issues to acquire and use EO data? |
| Do you use only free available data or data from private sources? Why? Which ones? |
| What do you expect of the main providers of EO data? Please comment based on your experience and knowledge in the field. |
| Have you ever visited INPE website to download any EO image? What do you think? |
| How would you rate the quality of the images available? Why? Please specify. |
| How likely are you to purchase a higher resolution image? |
| What are the major environmental and conservation issues in the country? |
| How could EO help to protect and monitor different landscapes? Any particular region or issue? |
| Based on the current political and economic situation in Brazil, please comment on institutional issues that affect your business or organization? How EO could improve your activity? |
| What are the main challenges to promote the use of EO data in Brazil? |
| To what extend is EO data being used for land use management in Brazil? |
| What is your view on the new forest code and the integration between agriculture and forestry? |
| Do you have any comments or further issues you would like to raise? Please offer your opinion on any not already mentioned in the interview. |

**Table A1.** *Cont.*

| Phase 2 (13 Questions) |
| --- |
| What type of EO data does your organization use? |
| For what purposes does your organization use EO data? Please indicate the name and use you give to the EO data. |
| Diagram of the system of how you use the EO data (Suppliers customers and use of EO information). Please draw a diagram with your ideas. |
| From where does your organization source the EO data? |
| Do you have in-house capability for image interpretation or do you need to contract services from an external provider? If so, which provider? |
| We have lots of different EO data (Terabytes of EO data archives). How do you think the user know how to select the best one? |
| Please outline how EO data are processed in your organization (steps involved, who does it). Please use the diagram to explain. |
| Are you restricted by type of data? For example, would high resolution with high frequency revisits help vs Landsat images, radar satellites can penetrate clouds and pick up topography? |
| What tools, training or capacity/capacity building are available for EO users? In what ways can these be enhanced to help improve EO data use? |
| How do you think your organization capacity associated with implementation of projects? Please draw a diagram with your ideas |
| How do you think different user's capabilities limits the implementation of projects? |
| Do you use indicators? Please give examples. |
| Do you see potential for greater use of EO data within your organization? |

## Appendix B

**Table A2.** Institutions and number of interviewees per institution in Phase 1 data collection.

| Location and Institutions (Phase 1) | Number of Interviews |
| --- | --- |
| Brasilia (Ministries and federal organizations) | 27 |
| Sao Jose dos Campos (INPE, National Institute for Space Research) | 8 |
| Sao Jose dos Campos (AGROTOOLS, Environmental consultancy company) | 1 |
| Campinas (EMBRAPA, Brazilian Agricultural Research Corporation) | 3 |
| Campinas (CEPAGRI, meteorology centre) | 1 |
| Curitiba (SIMEPAR, EMATER, ITCG, state and federal organizations) | 3 |
| Total interviews | 43 |

## Appendix C

**Table A3.** Institutions and number of interviewees per institution in Phase 2 data collection.

| Location and no. of Interviews | Institutions (Number of Interviewees per Institution) |
|---|---|
| Brasilia 17 | Brazilian Institute of Environment and Renewable Natural Resources IBAMA (3)<br>National Indian Foundation FUNAI (2)<br>National Food Supply Company CONAB (2)<br>Chico Mendes Institute for Biodiversity and Conservation ICMBIO (1)<br>Ministry of Environment MMA (2)<br>Brazilian Agricultural Research Corporation EMBRAPA (1)<br>Bank of Brazil Agribusiness department (1)<br>Brazilian Forest Services BFS (3)<br>Amazon Environmental Research Institute IPAM (1)<br>National Institute for Colonization and Agrarian Reform INCRA (1) |
| Cuiaba 19 | IBAMA Mato Grosso (2)<br>CONAB Mato Grosso (1)<br>ICMBIO Mato Grosso (1)<br>Bank of Brazil Agribusiness department (1)<br>INCRA Mato Grosso (2)<br>Federal University of Mato Grosso UFMT (2)<br>Federal Institute of Mato Grosso IFMT (1)<br>Agriculture Defense Institute of the state of Mato Grosso INDEA (1)<br>Association of Mato Grosso Breeders ACRIMAT (1)<br>Secretariat of Agriculture, Family Farming and Land Affairs SEAF (1)<br>Public attorney of Mato Grosso MP (1)<br>Federation of Agriculture and Livestock of Mato Grosso FAMATO (1)<br>Secretary of State for the Environment SEMA (1)<br>Mato Grosso Institute of Agricultural Economics IMEA (1)<br>Land Institute of Mato Grosso INTERMAT (1)<br>State Secretariat of Planning of Mato Grosso SEPLAN (1) |
| Rio Branco (Acre) 17 | ICMBIO Acre (1)<br>EMBRAPA Acre (1)<br>State Secretary for the Environment SEMA (2)<br>Central Unit of Georeferencing and Remote Sensing UCGEO (4)<br>Federal University of Acre UFAC (3)<br>Federal Institute of Acre IFAC (1)<br>Civil Defence of Acre (1)<br>INCRA Acre (1)<br>The Environmental Services Development Company CDSA (1)<br>Land Institute of Acre ITERACRE (1)<br>State Secretariat of Family Production and Rural Extension SEAPROF (1) |
| Total 53 | |

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
