# Peer review of "Information from Earth Observation for the Management of Sustainable Land Use and Land Cover in Brazil: An Analysis of User Needs"

_sustainability, doi:10.3390/su12020489_

Round 1

Reviewer 1 Report

The paper could be improved by attending to the following:

-consider placing Tables 1 and 2 in supplemental material or appendix. It is important for readers to have access to the questions and respondent types, but it is not essential to the text.

-the "conceptual model" should be modified by some terms.  It is a conceptual model of what? Problems faced by EO data producers?

-p. 12, lines 691 and 702: this is the part of the paper where sustainable land use and land cover are specifically mentioned. It is important to give these issues more attention because of the nature of the journal (Sustainability) and the title of the paper.  CAR needs more context, at minimum, so that readers could be convinced that it is part of sustainable land use planning.  Deforestation monitoring has a deep literature in Brazil and high significance.

-p. 17, line 1145: help readers understand the distance from Brasilia to Acre and the significance of the idea in this section of the paper.

-in the Discussion, the authors need to return to the findings of Rajao and colleagues.  How do the findings here advance those of Rajao's?

-p. 21, line 1480: I'm not sure the authors are using "conflicts of interest" in the sense they are meant.  Conflicts of interest normally mean that someone's actions in one area are compromised or possibly corrupt because of an interest that person has in a related domain.  I think the authors mean something different here, perhaps something like, "conflicts exist in the user domain because ... "

-section 5: provide specific conclusions regarding sustainable land use/cover, per the title and concerns of the journal.  Focusing on monitoring Amazonian deforestation and the CAR would improve the conclusion.

Reviewer 2 Report

The revised version of the paper “Information from Earth Observation for the management of sustainable land use and land 3 cover in Brazil: An analysis of user needs” better meets the criteria of a typical paper published in Sustainability journal.

The effort made by authors who have responded constructively to reviewers’ suggestions deserves the publication.

However, I have still found some inaccuracies requiring a further revision of the English.

In the following, some of the identified inaccuracies are reported:

First of all, there are several compound adjectives that could benefit from the presence of a dash between the two words to make the reading easier. For example: high-resolution (page 20 line 1373); I would like to see the Latin expression e.g. followed by comma as reported by most of international style guides. For example: page 2 line 211 (g.,) ; In different cases, several lines miss between one page and another; It is correct to insert a comma before the word etc (etcetera). See, for example, page 3 line 246; Page 1 line 22: multifaceted in state of multi-facetted; Page 1 line 28: land use/cover  in state of use and land cover; Page 2 line 187: in the sentence “it has been said that freely available EO are very important” I would add the word data after EO; I would rephrase the sentence at page 3 lines 277-278: “However, there remains a lack of research on the challenges faced by users when they try to access, use and process EO data “ in state of “However, there remains a lack of research on the challenges users confront when trying to access, use and process EO data”; I would rephrase the sentence at page 4 lines 303-304: “in supporting the management of sustainable land use and land cover (LULC) in Brazil” in state of “in supporting the management of sustainable land use and land use cover (LULC) in Brazil”; I would rephrase the sentence at page 4 line 311: “that needed to be asked” in state of  “that needed to asked” ; Page 4 line 318. focused in state of focussed; I would rephrase the sentence at page 4 lines 324-325: “all the interviewees’ names have been anonymized” in state of “all interviewee names have been anonymized” (possessive case); I would rephrase the sentence at page 4 lines 325-326: “The questions in each of the two phases of data gathering interviews are given in Table 1.” in state of  “The questions in each of the two phase of data gathering interviews are given in Table 1.” ; Page 6 line361: Table 2 should be substituted by Table 1; I would rephrase the sentence at page 7 line 384: “All the interview material of the Phases 1 and 2 was subjected to” in state of “All the Phase 1 and 2 interview material was subjected to”; I would rephrase the sentence at page 7 lines 387-388: “It is recognised that the each in-depth interview gathered data from one person at one time” in state of “It is recognised that the in-depth interviews each gathered data from one person at one time”; I would rephrase the sentence at page 8 lines 403-404: “although where the latter was mentioned by some interviewees those elements have been included” in state of 2 although where the latter was mentioned by some interviewees that elements has been included”; Page 9 line 473: Illustrates in state of illustrate; Pages 9-10 lines 476-496: repetition of the same sentence. Please remove one of them; I would rephrase the sentence at page 10 lines 517-519: “Between INPE and EMBRAPA we have about 300 people in different groups that are specialists in Earth Observation. EMBRAPA uses EO data for different applications in Agriculture, Forestry and the Environment” in state of “Between INPE and EMBRAPA we have about 300 people in different groups that are specialists in Earth Observation. EMBRAPA use EO data for different applications in Agriculture, Forestry and the Environment.” Page 11 Line 624: high in state of hight; Page 16 Line 1074: “the quantity and type of EO data available are not in itself” in state of “the quantity and type of EO data available is not in itself”; Please add full stop at the end of line 1084 (page 16); Page 17 Line 1137: “This aversion to information in government was only mentioned by only one interviewee” in state of “This aversion to information in government was only mentioned by the one interviewee”; Page 19 line 1310: “This research is the first one to take” in state of “This research is the first to take”; Page 19 line 1311: ‘in applying EO to help/facilitate sustainable management” in state of “in applying EO to help facilitate sustainable management”; Page 21 line 1466: perspectives in state of perspective; Add a full stop at the line 1645 (Page 22).

Author Response

This manuscript is a resubmission of an earlier submission. The following is a list of the peer review reports and author responses from that submission.

Round 1

Reviewer 1 Report

I think that the paper has a very poor structure both for design and used methods. Additionally, the use of an informal English, widely observed in the direct speech of the interviews, makes the reading difficult. I suggest to resubmit the paper, after substantial modifications in style and content (see some observations below), on a different type of journal such as the MDPI “Social Sciences”.

In the following, some suggestions to improve the formal quality of the paper:

1.      First of all, there are several compound adjectives that could benefit from the presence of a dash between the two words to make the reading easier. For example: semi-structured (row 20); end-users (row 25);

2.      There are several spaces to be eliminated or added within the manuscript (see, for example rows 44; 60);

3.      I would like to see the Latin expression e.g. followed by comma as reported by most of international style guides.

4.      Within the body of the manuscript, the reference to the figures 1 is missing. Furthermore, the exact and effective arrangement of tables and figures should be carefully evaluated. Possibly, tables and figures should be simply positioned as near as possible to the place where you first refer to them.

5.      I have noted some discrepancies between the results reported by the authors on the second analysis of the users profile of CBERS EO (Page 2, rows 66-70) and the results found in the reference n. 5 (Silva et al., 2009);

6.      Row 97: Capitol should be substituted for Capital;

7.      Page  4 row 148: complement should be substituted by complements;

8.      Page  7 row 285: understand should be substituted by understands;

9.      Page 7 row 300: people is plural in English!

10.  Page  7 row 323: the sentence “then the key point are” should be rewritten as “then the key point is”;

11.  Page 10 row 442: the sentence “The emerging cloud computing technologies provides new opportunities…” should be rewritten as “The emerging cloud computing technologies provide new opportunities…”;

12.  Page 10 rows 445-446: the sentence “Other private providers such as the Amazon Web Services and Planet Labs also provides cloud computing capabilities…” should be rewritten as “Other private providers such as the Amazon Web Services and Planet Labs also provide cloud computing capabilities…”.

Reviewer 3 Report

I enjoyed reading this paper very much. It fills an important gap in knowledge: who uses EO in Brazil, and why? What are challenges?  I felt as though the paper does a compelling job in advancing our knowledge about these issues.  Relevant work is cited.  Methods are mostly well described. Detailed qualitative results are presented--clearly, the authors obtained the trust of the respondents, to their credit!  However there are also some areas for improvement:

-authors could present the structure of the interviews. What were respondents asked?

-authors could present more information regarding the coding of the interviews.  It appears that coding, rather than content analysis (line 108), was performed. Analysis of the number of mentions of certain topics could help provide more detail in the overlaps (line 113).

-the English standard of the the narrative portion of the paper is high, but the translations from Portuguese to English should be revised. For example, line 153, line 173, line 226, etc.  Other aspects include 5,000 cities (194), which should be 5,000 municipalities (=municipios).  Original Portuguese could be used for the names of the organizations and agencies mentioned.

-consider using "human capital" instead of "capacity", eg line 366, or add a descriptor to capacity (eg what kind of capacity?).

-I'm not sure the paragraph between lines 436-451 adds much to the paper.

-I think a better case could be made for care around the term "sustainable land management" (line 468, 532), in the sense that the authors are taking at face value the idea that respondents are engaged in sustainability practices (but of course that is debatable!). It is possible, perhaps likely, that the respondents interviewed for this paper consider that they are using EO products for sustainability, but it does not seem that the paper tried to evaluate the specific aspects of sustainability that are represented because the focus was on land management. One example of this ellision between "sustainability" and land management is the abstract and Conclusion (line 540-551) emphasizes land management, but the discussion introduces "sustainable land management."